# Caregiver experience and perceived acceptability of a novel near point-of-care early infant HIV diagnostic test among caregivers enrolled in the PMTCT program, Myanmar: A qualitative study

Win Lei Yee[1], Kyu Kyu Than[1], Yasmin Mohamed[2,3], Hla Htay[1], Htay Htay Tin[4], Win Thein[4], Latt Latt Kyaw[4], Win Win Yee[4], Moe Myat Aye[4], Steven G. Badman[5], Andrew J. Vallely[5,6], Stanley Luchters[2,3,7,8]*, Angela Kelly-Hanku[5,6], on behalf of the AAMI study group[¶]

1 Burnet Institute, Yangon, Myanmar, 2 Burnet Institute, Melbourne, Australia, 3 School of Public Health and Preventive Medicine, Monash University, Melbourne, Australia, 4 National Health Laboratory, Ministry of Health and Sports, Yangon, Myanmar, 5 The Kirby Institute for Infection and Immunity in Society, UNSW Sydney, Sydney, Australia, 6 Sexual and Reproductive Health Unit, Papua New Guinea Institute of Medical Research, Goroka, Papua New Guinea, 7 Department of Population Health, Aga Khan University, Nairobi, Kenya, 8 Department of Public health and Primary Care, International Centre for Reproductive Health (ICRH), Ghent University, Ghent, Belgium

¶ Membership of the AAMI study group is provided in the Acknowledgments.
* stanley.luchters@aku.edu

**Data Availability Statement:** As ethics approval (through the Alfred Hospital HREC and in-country

## Abstract

### Background

The majority of HIV infection among children occurs through mother-to-child transmission. HIV exposed infants are recommended to have virological testing at birth or 4–6 weeks of age but challenges with centralized laboratory-based testing in Myanmar result in low testing rates and delays in result communication and treatment initiation. Decentralized point-of-care (POC) testing when integrated in prevention of mother-to-child transmission of HIV (PMTCT) services, can be an alternative to increase coverage of early infant diagnosis (EID) and timely engagement in HIV treatment and care.

### Aim

This paper aims to explore experiences of caregivers of HIV-exposed infants enrolled in the PMTCT program in Myanmar and the perceived acceptability of point-of-care EID testing compared to conventional centralised laboratory-based testing.

### Methods

This is a sub-study of the cluster randomised controlled stepped-wedge trial (Trial registration number: ACTRN12616000734460) that assessed the impact of near POC EID testing using Xpert HIV-1 Qual assay in four public hospitals in Myanmar. Caregivers of infants who

ethics committees in Myanmar) was subject to the data being securely stored on a password protected database, with access restricted to authorized members of the research staff, we are unable to deposit the data in a public repository. Also, due to the in-depth nature of the data and the small communities where data collection took place, the data transcripts could contain potentially identifiable quotations - and potentially cause shame or embarrassment, especially in settings where there is stigma associated with HIV/AIDS. Given the ethical restrictions on sharing data publicly, data sharing will be made possible on request via the Alfred Hospital HREC (research@alfred.org.au, project number 500/14) and the Department of Medical Research IRB (ercdmr2015@gmail.com, approval number Ethics/DMR/2016/115).

**Funding:** Support was provided by the National Health and Medical Research Council of Australia (NHMRC) through Project grant GNT1063725, and Career Development Fellowship to S Luchters. The fundershad no role in study design, data collection and analysis, decision to publish, or preparation of the manuscript.

**Competing interests:** The authors have declared that no competing interests exist.

were enrolled in the intervention phase of the main study, had been tested with both Xpert and standard of care tests and had received the results were eligible for this qualitative study. Semi-structured interviews were conducted with 23 caregivers. Interviews were audio recorded, transcribed verbatim and translated into English. Thematic data analysis was undertaken using NVivo 12 Software (QSR International).

## Results

The majority of caregivers were satisfied with the quality of care provided by PMTCT services. However, they encountered social and financial access barriers to attend the PMTCT clinic regularly. Mothers had concerns about community stigma from the disclosure of their HIV status and the potential consequences for their infants. While medical care at the PMTCT clinics was free, caregivers sometimes experienced financial difficulties associated with out-of-pocket expenses for childbirth and transportation. Some caregivers had to choose not to attend work (impacting their income) or the adult antiretroviral clinic in order to attend the paediatric PMTCT clinic appointment. The acceptability of the Xpert testing process was high among the caregiver participants and more than half received the Xpert result on the same day as testing. Short turnaround time of the near POC EID testing enabled the caregivers to find out their infants' HIV status quicker, thereby shortening the stressful waiting time for results.

## Conclusion

Our study identified important access challenges facing caregivers of HIV exposed infants and high acceptability of near POC EID testing. Improving the retention rate in the PMTCT and EID programs necessitates careful attention of program managers and policy makers to these challenges, and POC EID represents a potential solution.

## Introduction

In 2018, 1.7 million children aged 0–14 years were living with HIV [1] and around 160,000 children (0–14 years) were newly HIV infected globally [2]. In the absence of antiretroviral treatment (ART), half of all HIV positive children die before two years [3–5]. However, ART initiation within the first three months of life can reduce HIV related mortality by 75% [3]. Despite this, there is suboptimal access to ART among HIV-infected children (<15 years) with a global estimate of a mere 54% receiving such lifesaving treatment in 2018 [1]. In addition to antiretroviral prophylaxis during the first six weeks after delivery, early diagnosis of HIV among exposed infants and initiation of ART among infected infants are the single two most effective ways to reduce neonatal morbidity and mortality among this population. While WHO guidelines recommend virological testing for all HIV exposed infants as early as at birth and/or at 4–6 weeks of age [6, 7], little more than half (54%) of exposed newborns globally are tested within two months after birth [4]. In many settings, poor rates of virological testing are due to the need for centralized laboratory-based testing with highly trained staff, often leading to a long turnaround time of results. In these conditions, opportunities to initiate early lifesaving ART are missed. In the meantime, HIV positive women and infants are lost to follow-up, untreated infants progress quickly to develop opportunistic infections associated with AIDS

(and treatment—if started—is based on presumptive clinical diagnosis) and too many die [4, 5, 8, 9]. In Myanmar, it is estimated that in 2018, there were approximately 9,800 (8,400–11,000) HIV infected children aged 0–14 years [10]. HIV prevalence among women attending antenatal care varies across the country from as low as 1% to 3% with the highest number reported in major hospitals in Yangon, the former national capital, and other cities including Mandalay, Tanintharyi and Myitkyina [11]. Significant progress has been made in the prevention of mother-to-child transmission of HIV infection (PMTCT) including the integration of PMTCT programs into 38 antenatal care services in public hospitals [11].

Mother-to-child transmission of HIV occurs during pregnancy, labour, delivery and breast-feeding, and thus, WHO recommends to integrate prevention of mother-to-child transmission services with maternal and child health services and HIV treatment and care programs [12, 13]. In 2015, 3,923 HIV positive mothers in Myanmar received ART and 97% of all HIV exposed infants received nevirapine prophylaxis. However, among the 2,239 babies delivered by HIV positive mothers in 2015, only one-third (36%) received an HIV test within two months of age as recommended by WHO [11]. This very low rate of early infant diagnosis (EID) of HIV is an area of concern identified in Myanmar's National HIV Strategic Plan, suggesting a need to establish a cascade of comprehensive PMTCT services that includes prenatal HIV testing and ART for mothers, paediatric ART prophylaxis and early HIV testing for infants, and functioning case management system [11]. In order to address the inadequate uptake of EID, significant delays in returning the results and timely initiation of ART in resource-limited settings, a multinational randomized controlled clinical study on point-of-care (POC) Xpert® HIV-1 Qual test (Cepheid, Sunnyvale, CA, USA) was conducted in Myanmar and Papua New Guinea (PNG) from 2016 to 2018. This was the first time that POC EID testing had been offered in either country, hence understanding the perspectives of caregivers and health care providers towards this new diagnostic technology is critical if it is to be scaled up and used in the future. In this paper, we focus on implementation aspects in Myanmar, and explore the perspectives of caregivers towards PMTCT services more generally, and the acceptability of the implementation of near POC EID into routine services.

## Methods

### Study setting

A cluster randomised controlled stepped-wedge trial (Trial registration number: ACTRN12616000734460) that assessed the impact of near POC EID testing using Xpert HIV-1 Qual assay was conducted in four major public hospitals in the Yangon Region of Myanmar from October 2016 to June 2018. These study hospitals had routine antenatal care clinics with integrated PMTCT services such as HIV testing with pre-test and post-test counselling. Pregnant mothers who tested HIV positive were referred to ART centers for treatment initiation. Two of our study hospitals (Thingangyun General Hospital and Thanlyin General Hospital) were attached to an ART center whereas the other two hospitals (South Okkalapa Women's and Children's Hospital and Central Women's Hospital) referred the HIV positive mothers to one of the ART centers outside the hospital in Yangon during the study period. Regardless of where the mothers chose to deliver, antenatal services and infant care after delivery until 18 months of age were provided through the PMTCT clinics of the hospital.

In collaboration with the National Health Laboratory (NHL) and National AIDS Program, Xpert® HIV-1 Qual assay (Cepheid, Sunnyvale, CA, USA) was introduced to these hospitals in a randomised step-wise manner to conduct EID testing so that all four hospitals received the intervention by the end of the study. A two-module device was installed at the laboratory of the hospitals and assigned laboratory technicians were trained on operation procedures and

quality assurance before the start of the study and during the transition from the control to intervention phase of the study. These trained laboratory technicians performed the test and communicated the results back to the treating medical doctors or paediatricians. Following WHO and national guidelines [14, 15]), HIV exposed infants who were enrolled during the control phase of the study had standard of care HIV testing which involved collecting dried blood spots (DBS) from the heel or big toe of infants between 4–6 weeks of their age at the hospital laboratory. The DBS samples were sent to NHL to test with Abbott (Abbott, Abbott Park, Illinois, USA) and the results were delivered back to the hospitals. Infants enrolled during the intervention phase of the study had both the standard of care test and the Xpert test.

## Study design

The data of this paper are drawn from a qualitative sub-study of the larger cluster-randomized controlled stepped-wedged trial. The caregivers of infants enrolled in the main study were administered a questionnaire regarding their HIV and ART history, breastfeeding practices and infant care, during their enrolment visit and the subsequent visits for the testing and the result. In order to gain an in-depth understanding of caregivers' experiences with POC EID compared with conventional testing methods at the centralised laboratory (standard of care), qualitative semi-structured one-to-one interviews were conducted with the caregivers of infants who participated in the intervention phase near the end of the main study.

## Study participants

A convenience sample of caregivers were selected for participation in this qualitative acceptability sub-study. The caregivers attending the PMTCT follow-up appointment of their infants were identified and checked for their eligibility. Eligibility included caregivers with infants of the intervention phase of the study who had undergone the EID and received both Xpert and the standard of care results, those who accompanied the infants in testing and results visits and who were willing to provide informed consent for the interview. Caregivers of infants who met the eligibility criteria and experienced repeated blood withdrawal, delayed testing and result return were also invited purposively into the study. Pseudonyms are used when presenting their quotes in this paper in order to protect their confidentiality.

## Study procedures

Two experienced female qualitative researchers from Myanmar who were independent of the larger field trial conducted the interviews separately. A semi-structured interview guide was designed, piloted among the study staff and revised, and then administered for data generation. The interview guide focused on caregivers' perceptions of the quality of the services, barriers to accessing PMTCT clinic and laboratory tests for their child, their understanding of the study procedures, experiences of their child's EID testing, and acceptability and validity of the same day Xpert test result. All 23 caregivers who were interested to participate were explained the purpose of the study and the study procedures including risks and benefits. Written informed consent was obtained from all participants. Interviews took between 45 to 60 minutes. Most of the interviews took place in a private room at the hospital; however, in some hospitals where a room could not be arranged, the interviews happened at the outpatient clinic or somewhere in the hospital compound with limited privacy. Data collection continued until data saturation was reached and no new themes emerged. Interviews were audio recorded, transcribed verbatim and translated into English by the researchers who conducted the interviews. Translated interviews were cross checked for quality assurance purposes.

## Data analysis

A third researcher, the first author of this paper, undertook the qualitative data analysis of the caregiver interviews. Using NVivo 12 Software (QSR International), a deductive analysis was undertaken based on a predesigned code book based on the interview guide. Following the deductive analysis, the same researcher conducted an inductive one where new and emerging issues could be coded for analysis. Transcripts were read and re-read repeatedly to be familiar with the contents and emerging themes. Standards for reporting qualitative research (SRQR) [16] was applied when reporting the findings in this paper.

## Ethical approval

Ethical approval was obtained from Ethical Review Committee of Department of Medical Research (Ethics/DMR/2016/115) in Myanmar and in Australia by the Alfred Hospital Ethics Committee (Project 500/14).

# Results

## Characteristics of participants

A total of 23 caregivers were interviewed with an average age of 29 years. Excluding one aunt, other caregivers interviewed were the birth mothers of the exposed infants enrolled. Just under half (11/23) of mothers had been diagnosed with HIV prior to their most recent pregnancy and the others (12/23) were diagnosed as part of routine antenatal care during the most recent pregnancy, and were therefore relatively newly diagnosed with no experience of standard early infant diagnosis testing. The majority of mothers were delivered by Caesarean section (19/23 mothers) and reported having bottle fed their infants (16/23 infants) by six weeks. In Myanmar, Caesarean section is the preferred mode of delivery by most of the obstetricians for mothers with known HIV infection giving birth at the hospital. Of the infant-caregiver pairs included in this qualitative study, all infants tested HIV negative when tested between 4 and 6 weeks.

## Key findings

Using thematic analysis of the 23 caregiver interviews, we identified two main themes: (1) the accessibility of PMTCT services, and; (2) the acceptability of Xpert for diagnosing the infants in their care. The first issue is important as to date there is no published data in Myanmar on the experiences of women in PMTCT programs and this is intimately related to the second issue on GeneXpert testing. Unless the data included the issues of the one aunt in the study, we will refer to mothers.

## Accessibility of PMTCT services

**Social factors.**   Fear of unwanted disclosure affected the experience of attending PMTCT clinics, even in large urban areas. Concerned that they would be seen by others including their neighbours and extended family and that in turn they would be discriminated against because of their status, many women lived in fear during the time of the journey and of attending the PMTCT clinic.

*If I have this kind of disease, I would be discriminated among my community as this is regarded as unacceptable. It is hard to identify who has the disease in urban community unlike our village community where news spread easily. It is really embarrassing*

(Ma A, mother, 28 years).

*She [the mother's neighbour] accompanied me at the first time to get medicine for my eldest daughter. I am scared of being discriminated if the neighbourhood found out*

(Ma B, mother, 32 years).

Mothers made informed and calculated choices about which health facility they attended. This choice was made on the basis of a belief that their confidentiality and anonymity would be ensured and they would then avoid potential stigma and discrimination.

*Financial issue is not such a burden on me, but the social issue is. When I came here, I met with my neighbour...I don't want to let anyone know that I have the disease [HIV], except my family... There is an Urban Health Center in [name of place removed] where I can get all the care and child immunization. I was supposed to be referred there for my child's immunization, but I asked [the doctor] not to as I know many people there and that is why I came here [the hospital] intentionally*

(Ma C, mother, 32 years).

**Transportation factors.**   Caregivers expressed concern about the distance from home to the heath facility where PMTCT services are provided.

*Last visit, I was asked if it is convenient for me if they refer me to Mingalardone specialist hospital, I replied 'No' because it might be difficult to go there with my small baby. So they sent me to Thanlyin Hospital which is rather closer to my residence. It takes about half to one-hour ride to Thanlyin Hospital from Khayan so they gave me a referral letter*

(Ma A, mother, 28 years).

*Actually, coming to the follow up is hard for me as it is too far [from home]. But I tried my best to make it because it is for my baby*

(Ma G, mother, 34 years).

Although the caregivers attended the PMTCT clinics as required, the cost of transportation to and from the clinic was a financial challenge.

*When the baby was very young, we took a taxi instead of the motorbike. It cost around 10,000 kyat [USD 70] for the visit. It was a bit costly and I cannot afford more*

(Ma D, mother, 19 years).

*The transportation cost alone is at least 30,000 kyat [USD 20]. I have to take the taxi because there is no direct bus running to this side of the town*

(Ma A, mother, 28 years).

**Financial factors.**   Although services for childbirth and attending PMTCT clinics at the study hospitals were free of charge, HIV positive mothers during childbirth and after delivery requiring regular attendance to PMTCT clinic for their infants faced financial pressures in multiple ways. Costs during hospital stays and for accompanying persons were commonly reported financial difficulties alongside the transportation costs described above.

Some mothers came to the clinic accompanied by someone such as her husband or an extended family member, which cost them further money to cover the additional food and transportation.

*I brought someone for the day of blood taking. If not so, I came by myself. If I asked to someone to accompany me, it costs me extra for meals and transportation*

(Ma J, mother, 40 years).

*I am not able to come alone with the baby. I need someone to accompany me. It is a bit far to come here and the baby is breastfeeding. . . I have to hold the umbrella in one hand and carry a bag. I felt it is better if someone accompanies me*

(Ma G, mother, 34 years).

On the other hand, a few women with a lack of family support for childcare struggled to balance the demands of paid formal employment and needing to attend the clinic. In the absence of family support these women had to forgo paid days at work in order to ensure their children were at the clinic.

*When I came here [the clinic], I had to take leave from my work, so I did not get paid for the day*

(Ma K, mother, 34 years).

*He [the father] had to take one day leave from his work because he needs to come along with me. It is important for our baby*

(Ma L, mother, 26 years).

These financial strains coupled with meeting the basic needs of the family including the home they live in.

*Since I gave birth in the rainy season, the roof of our house broke due to heavy rain. So that I couldn't bring my child for follow up*

(Ma M, mother, 32 years).

There was an occasion when the hospital staff wanted money and demanded under-the-table payment from mothers delivering at the hospital. One of the mothers detailed her experience about monetary requests made by hospital staff.

*The doorman was not behaving well as he, it seems, wanted a tip. . ..and some nurses were really bad. . .. They asked me money when I came out of the operation theater. A red staff [nurse]) came to us and told me that: "if you guys do not treat us well, I won't treat you well either". That means she was indirectly asking for money*

(Ma H, mother, 32 years).

**Time availability factors.** In the absence of combined services for mothers and HIV exposed infants, many mothers expressed the difficulties to come to the PMTCT clinic if their own appointment at the adult ART centre conflicted with the appointment of their child.

*Sometimes the appointment day for my baby coincided with my appointment day to go to Insein hospital for my treatment. If so, it was difficult to manage for both*

(Ma H, mother, 32 years).

*Sometimes, I have to go and take my medication at another hospital. If the baby clinic's day coincides with my clinic day, it is not convenient for me*

(Ma I, mother, 32 years).

**Quality of care.** Many mothers were satisfied with the services at the PMTCT clinic and reported the good quality of the care provided by health care workers at the study hospitals.

*Everything is fine. I am satisfied. The hospital staff took care of me in the delivery room patiently. The nurse showed up very quickly when we requested them*

(Ma P, mother, 32 years).

However, some mothers expected better communication and courteous behaviour from hospital staff when dealing with them for their disease and infant care.

*I just want to be treated friendly and warmly. I am satisfied if I got treated warmly, and just friendly. We are infected persons so our feelings are down and depressed. . . Only thing I want to improve is the communication of the staff, especially the nurses*

(Ma I, mother, 32 years).

The majority of the participants perceived the attitudes of health care workers as supportive and non-discriminatory.

*Because we have this disease, they help us a lot. They do not blame us for getting this disease. Just tell us to take medicine regularly and take care of health ourselves*

(MH, mother, 26 years).

*I am satisfied with all the staff and services because they don't discriminate me*

(Ma C, mother, 32 years).

However, a few mothers expressed feeling stigmatised and discriminated against by the health care workers in the PMTCT services.

*I can understand from their eyes contact to each other. I felt inferior when they wore gloves. I know that I am a dangerous patient when I come for medical check-up because on the paper it has the word 'positive'*

(Ma B, mother, 32 years).

Having a chance to engage with other HIV positive mothers at the PMTCT clinic increased the clinic's appeal. The mothers felt a friendly environment from meeting with other HIV positive mothers at the PMTCT clinic. An opportunity of having a conversation with their peers and exchanging their own experiences that could not be disclosed to non-infected people while waiting at the clinic created an emotional outlet for the mothers.

*The hospital service is really good. Nothing to complain. . .The best thing here is I met with other infected mothers. We chatted and shared our experience each other. Here, we, among the same experienced people, can talk frankly about our babies. We cannot do the same talks in our neighborhood*

(Ma F, mother, 36 years).

*Nobody can identify who was [HIV] infected or not because many mothers come and seek care here. Besides, an infected mother feel that she was not alone, because she can meet with similar mothers who came and seek care to prevent transmission to their babies. . .*

(Ma J, mother, 40 years).

## Acceptability of POC EID

**Experiences of receiving Xpert test result.** Despite being designed as a test with the availability of same day results to overcome the turnaround time of conventional laboratory-based EID testing, only 14 of the 23 caregivers reported having been informed of their baby's HIV test results on the same day. Results were available to the 14 caregivers because they waited for results, returned later in the day or had a relative wait and receive the results on their behalf:

*I was so happy that the result would be given immediately. The staff told me I could either wait or come back later. So, I decided to wait to get the result*

(MT, mother, 26 years).

*After taking the blood sample, I asked my sister to wait for the result, and I went back home. On that day, there was a funeral of my grandma. . . I let my sister take the result*

(MN, mother, 22 years).

However, others (9/23) came back to get the result on a different day as they lived nearby the hospital or the laboratory could not provide the result on the same day. In this case, they were given an appointment on the following clinic day or the day the doctors/paediatricians were available.

*Yes, three days. The blood sample taking was done on Friday and I got the result on Tuesday*

(Ma G, mother, 34 years).

*There were two patients ahead of me and the blood tests were done in a hurry as the lab will be closed at 2pm. They asked us to come after three days*

(Ma I, mother, 32 years).

The mothers also described their experiences of getting standard of care test results that took longer than the Xpert test result.

*For DBS testing, although I was told it would take one month, in reality, it took more than two months. It was nearly three months' time [to get DBS result]*

(Ma H, mother, 32 years).

*We'd get the first result immediately [Xpert test]. We'd get another result after about one and a half months. They said like that. I said, 'It's okay'. After one and a half*

*months I came to get the result [DBS routine test] and they said the result was not received yet*

(Ma M, mother, 32 years).

**Perceptions of the POC Xpert test.** One of the participants was unclear about Xpert result and the doctor wanted her to wait for the result of the standard of care test, showing the doctor's limited trust in the Xpert test.

*I got the result [Xpert test]. When I went to the child care unit and gave them the result, I was not told whether the child is positive or negative. But they told me that it will be surer only when another result [DBS standard test] comes out*

(Ma M, mother, 32 years).

Many mothers expressed positive attitudes towards receiving the result the same day as testing. They became aware of the need to undergo the testing again at 9 months and 18 months of the age at the same time and one of the mothers expressed her enthusiasm to come back for these tests.

*The benefit of this test is that I feel delighted and happy as I knew the result early. And I have hope for what to do in the future as the baby does not have the infection*

(Ma N, mother, 28 years)

*They told me that the result is negative. However, we had to test it again after 9 months of age. They can be sure [of not infected] after doing test on one-and-a-half-year of age*

(Ma O, mother, 18 years).

*It's good. I like it. . .I was really over joyful. I even liked to roll over and jump with joy. . . . I would come back to test again at her age of 9 months as I worried that unfortunately she would be infected*

(Ma F, mother, 36 years).

The mothers perceived that the Xpert test has advantages in testing procedures and giving out the result in a short duration of time, thereby reducing their worries and anxieties.

*It was fine. I was just pleased to know the result straightway. I was very excited to know it. It was good for me because I didn't need to wonder around here and there to get the test and wait for some time*

(Ma L, mother, 27 years).

*The benefits are more. It makes me relieve. If it takes long, I will be worrying for longer time. The earlier the better*

(Ma H, mother, 32 years).

## Discussion

The findings of this study describe the experiences of HIV positive mothers with PMTCT services and the acceptability of novel same day EID testing in Myanmar. Social stigma and

financial factors were apparent barriers for mothers accessing PMTCT clinics; however, the good quality of PMTCT services and the friendly environment enhanced their access to the clinic. The caregivers found that using the Xpert platform was highly acceptable because of the quick return of the results.

Retention of HIV positive mothers and their infants in PMTCT programs is critical in early infant HIV diagnosis and treatment. Although PMTCT clinics provided services mostly free of charge, women often dealt with indirect costs, such as transportation, absenteeism from work and extra expenses for their companions, which could predispose them to be loss to follow-up. Though this finding is not generalizable to the population at large, it could help to explain the reasons for loss to follow-up of HIV positive mothers from the PMTCT program. In a cohort study carried out in Mandalay region of Myanmar, among 678 pregnant women enrolled in the PMTCT program, 2.8% were lost to follow-up before delivery and 13% after delivery while 99 out of 607 HIV exposed infants (16%) became lost to follow-up before the final diagnosis of HIV [17]. An earlier study conducted in Uganda on the delay in ART initiation among infants also demonstrated the transportation cost being one of the reasons of late attendance of the children to the clinic [18]. Similarly, in Uganda, high transportation cost and long distance to ART clinics have been the barriers for HIV positive mothers to access and adhere to ART [19]. In our study, the participants were able to attend clinic appointments with their children, although, the fact that we interviewed caregivers already at the clinic likely influenced this finding. The accessibility challenges identified by our participants highlight a need to adapt the clinic hours and appointment scheduling to meet the needs of the caregivers to lessen their burden and improve their attendance at the clinic.

Many of the caregivers were satisfied with the PMTCT services and perceived the behaviour of health care staff as very positive. They largely felt that they were treated with respect and given more care than other patients. However, a few of the mothers experienced concerning interactions with staff and felt stigmatised. The communication problems, coming out of the power imbalance between providers and patients, occurs in many healthcare settings [20–22]. In studies conducted in Tanzania and Uganda, discrimination and disrespectful behaviour of the providers have reportedly influenced the adherence to ART and other PMTCT services [19, 23]. It is vital for health workers dealing with HIV positive mothers to be aware of their feeling of inferiority resulting from their infection, persistent stigmatization and their socio-economic status [23] and adopt respectful treatment for building trust and provider-patient relationship [24].

The literature shows that community attitudes and stigma against HIV infection prevents many people with HIV infection, including women and children, from accessing healthcare services. In our study, the existence of community stigma and discrimination against HIV infection became apparent through women's fear of their HIV status being known to their neighbours. The mother participants had concerns that society would stigmatize and discriminate against their child because of HIV infection. It is similar in South Africa where people living with HIV infection faced stigma if their child was found to be HIV infected [25] and experienced degrading behaviours that affected their access to HIV testing and care [26, 27]. Stigma and fear influenced the health seeking behaviour of caregivers [18] and there is evidence that suggested a link between reduced uptake of HIV testing and treatment of mothers and infants, and fear of their HIV status being known and subsequent stigma [28, 29]. As women living in fear of stigma are likely to use health services far from their residence, measures for community stigma reduction and linkage to social support services would increase their access to PMTCT services.

Delivering integrated HIV treatment and care services has the benefit of improving the access of HIV positive mothers and their infants to PMTCT services. Mothers enrolled in the

PMTCT program at the study sites who were taking ART from other ART centres sometimes encountered schedule conflicts and needed to choose between a PMTCT appointment for their child and their own ART appointment. In this sense, integration of ART centres into the PMTCT cascade could retain mothers and HIV exposed children in the program, without requiring the mothers to go to a different health facility for their own HIV care and treatment. A study at primary healthcare facilities in Mozambique found that one-stop integrated consultation for mothers and their HIV-exposed infants led to more efficient service delivery than separate consultations though there were structural limitations and little evidence of a significant improvement in their follow-up attendance [30]. A cluster randomised controlled study in Nigeria also showed that integrated PMTCT care engaging male partners and community improved the retention of mother and infant pairs in the program [31]. The findings of this study also indicated that supportive family members enabled mothers to receive early diagnosis of their infant, whereas negative societal attitudes towards HIV infected persons are likely to prevent mothers from accessing PMTCT services because of fear and shame.

The Xpert platform brought direct benefits to mothers by reducing their waiting time for the result and consequently, relieving their anxieties from waiting to know their child's HIV status. Knowing the test result of their infant is an emotionally tiring process for mothers, especially when being informed about their child's positive result [32]. In our study, the caregivers preferred a quick result communication so as to avoid stressful moments of waiting for their child's result. They appreciated the less complex testing procedures and short turnaround time of the novel point-of-care test.

While Xpert testing is meant to ensure timely diagnosis and initiation of treatment as needed, the women in this study highlight that this is not always the case. This was due to operational issues of the device including: the number of modules per machine; the location of the device; the person operating the machine, and; the operating hours of the PMTCT clinic. The PMTCT clinic of three of the study sites ran in the afternoon and the testing was done at the laboratory, which is relatively far from the clinic. Normally the clinics had a low caseload and the laboratory staff could perform the Xpert EID test and provide the results on the same day. Occasionally, the delays occurred when the number of tests exceeded the capacity of the two-module Xpert device or samples were received just before the laboratory was closed. In order to enable efficient testing and timely result return, there should be an appropriate adjustment of time between the laboratory and the PMTCT clinic, and the use of the GeneXpert system with proper module configuration depending on the case load of the health facility. This requires sufficient human resource and healthcare infrastructure with careful task allocation among the healthcare cadres.

Efforts have been made by several stakeholders in Myanmar to increase the accessibility of HIV exposed infants to timely testing and treatment in order to reduce HIV related infant mortality rate. Considering the need of complex laboratory settings and skilful technicians for centralised laboratory-based EID testing and its lengthy turnaround time [33], decentralisation to health facilities operating a PMTCT program would likely be an effective and feasible approach [34, 35]. Our findings suggest the favourable environment of the PMTCT program and high acceptability of the same day testing by the caregivers. Given that the delayed receipt of testing results impacts on retention rate in EID programs as well as early treatment initiation [36], Xpert EID could be a potential solution to improve EID coverage in Myanmar.

## Limitations of the study

There was only one infant who tested positive for HIV infection in the study and its caregiver could not be contacted for the interview. More than half of the caregivers were mothers who

had no previous experience with EID testing prior to our study and thus, their responses were limited to their current experiences with the standard of care test and POC test. The study was conducted in tertiary level health facilities in Yangon and hence, experiences could be different if the Xpert platform was to be established in smaller health facilities or in more rural settings.

## Conclusion

Caregivers of HIV exposed infants were satisfied with services provided by the PMTCT program, but faced financial, logistical and social barriers to accessing these services. They found the same day Xpert EID test to be favourable, emphasizing the simplicity of testing procedures and particularly the quick result turnaround time. Strengthening EID testing and treatment will require consideration for an integrated and inclusive PMTCT program for mother-infant pairs and a decentralised testing approach.

## Supporting information

**S1 File. "Accelerating HIV testing and ART initiation among infants (AAMI)" study group.**
(DOCX)

**S2 File. Interview guide for caregivers (English).**
(DOCX)

**S3 File. Interview guide for caregivers (Myanmar).**
(DOCX)

## Acknowledgments

The authors gratefully acknowledge the contribution to this work of the Victorian Operational Infrastructure Support Program received by the Burnet Institute. Cepheid provided the two-cartridge GeneXpert machines free of charge for study purposes. The authors also would like to acknowledge the women and children who agreed to participate in the study and the research team who contributed to this work.

The following are members of AAMI (Accelerating HIV testing and ART initiation aMong Infants) study group: Stanley Luchters (Principal Investigator stanley.luchters@aku.edu), Suzanne Crowe, Mark Stoové, David Anderson, Claire Nightingale, Paul Agius, Yasmin Mohamed from Burnet Institute, Melbourne, Australia; Hla Htay, Win Lei Yee (first author winlei.yee@burnet.edu.au) from Burnet Institute, Yangon, Myanmar; Angela Kelly-Hanku, Andrew Vallely, Steven Badman from The Kirby Institute for Infection and Immunity in Society, University of New South Wales, Sydney, Australia; Zure Kombati from Mt Hagen General Hospital, Mt Hagen, Papua New Guinea; Tin Maung Zaw from National AIDS/STD Control Program, Ministry of Health and Sports, Myanmar; Xiang-Sheng Chen from National Center for STD Control, Nanjing, China; Htay Htay Tin, Win Thein, Latt Latt Kyaw from National Health Laboratory, Ministry of Health and Sports, Myanmar; Janet Gare, Selina Silim from Papua New Guinea Institute of Medical Research.

## Author Contributions

**Conceptualization:** Stanley Luchters.

**Formal analysis:** Win Lei Yee.

**Funding acquisition:** Stanley Luchters.

**Investigation:** Steven G. Badman.

**Methodology:** Kyu Kyu Than, Steven G. Badman, Andrew J. Vallely, Stanley Luchters, Angela Kelly-Hanku.

**Project administration:** Win Lei Yee, Yasmin Mohamed, Hla Htay.

**Supervision:** Yasmin Mohamed, Hla Htay, Htay Htay Tin, Win Thein, Latt Latt Kyaw, Win Win Yee, Moe Myat Aye, Stanley Luchters, Angela Kelly-Hanku.

**Writing – review & editing:** Win Lei Yee, Kyu Kyu Than, Yasmin Mohamed, Hla Htay, Htay Htay Tin, Win Thein, Latt Latt Kyaw, Win Win Yee, Moe Myat Aye, Steven G. Badman, Andrew J. Vallely, Stanley Luchters, Angela Kelly-Hanku.

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
