## [Decision Letter · Decision Letter 0]

18 Sep 2020

PONE-D-20-22119

Prevention of mother-to-child transmission of HIV in Myanmar and the acceptability of a novel near point-of-care early infant HIV diagnostic test among caregivers: a qualitative study

PLOS ONE

Dear Dr. Lei Tee,

Thank you for submitting your manuscript to PLOS ONE. After careful consideration, we feel that it has merit but does not fully meet PLOS ONE’s publication criteria as it currently stands. Therefore, we invite you to submit a revised version of the manuscript that addresses the points raised during the review process.

We look forward to receiving your revised manuscript.

Kind regards,

Claudia Marotta

Academic Editor

PLOS ONE

Journal Requirements:

2. Please address the following:

- Please include additional information regarding the interview guide used in the study and ensure that you have provided sufficient details that others could replicate the analyses. For instance, if you developed a guide as part of this study and it is not under a copyright more restrictive than CC-BY, please include a copy, in both the original language and English, as Supporting Information. In addition, please provide further details of the pilot testing of this tool.

- Please ensure the clinical trial this is related to is written correctly as "Australian and New Zealand Clinical Trials Registry, number 12616000734460" could not be found.

3.We note that you have indicated that data from this study are available upon request. PLOS only allows data to be available upon request if there are legal or ethical restrictions on sharing data publicly. For information on unacceptable data access restrictions, please see http://journals.plos.org/plosone/s/data-availability#loc-unacceptable-data-access-restrictions.

4. One of the noted authors is a group or consortium [AAMI study group]. In addition to naming the author group, please list the individual authors and affiliations within this group in the acknowledgments section of your manuscript. Please also indicate clearly a lead author for this group along with a contact email address.

Additional Editor Comments (if provided):

Dear Authors,

follow reviewer suggestions to improve your very interesting paper.

Reviewers' comments:

Reviewer's Responses to Questions

**Comments to the Author**

1. Is the manuscript technically sound, and do the data support the conclusions?

Reviewer #1: Yes

Reviewer #2: Partly

2. Has the statistical analysis been performed appropriately and rigorously? 

Reviewer #1: Yes

Reviewer #2: N/A

3. Have the authors made all data underlying the findings in their manuscript fully available?

Reviewer #1: Yes

Reviewer #2: Yes

4. Is the manuscript presented in an intelligible fashion and written in standard English?

Reviewer #1: Yes

Reviewer #2: Yes

5. Review Comments to the Author

Reviewer #1: I read this interesting paper with pleasure. The idea research is valid nad also appreciate the field of research and the setting.

Only some suggestions:

1. Introduction: remove line 102, 110, 122. You can add the concept of child at risk (see and cite doi:10.3390/ijerph15071350) to underline how they are most vulnerable group and need also scientific attention

2. Methods: delete line 147. No other suggestions

3. Results: clear. It is a qualitative study and it is diffult to do a tables

4. Discussion: well done. Delete line 494, 513, 525, 538, 552, 560,570, 586. If you can compare your results with other experice to improve aderehence (doi:10.1186/s12889-018-5646-8 and doi:10.1186/1472-6963-13-207)

Reviewer #2: The topic is very important as the new POC EID test obviously lessen the waiting time of the caregivers to hear their infant's test result. However, this study has a methodological and coherence problem. First, the type of study should be clear. If this study is a comparative study, then there should be two distinct groups and measurement variables need to be presented comparatively to each group. For example, under the result section(L64-65), it says “the acceptability of the Xpert testing process was high among the caregivers …” but is not clear which group has high accessibility. I understand, there was a randomized control trial study. If you think, the parent study is helpful for the current study, the methodology of the main study should explicitly and separately explain. Similarly, the methods for the current qualitative study must be explained. Confusions with the current and previous studies must be cleared.

You can also find the detail comments in the uploaded file.

6. PLOS authors have the option to publish the peer review history of their article (what does this mean?). If published, this will include your full peer review and any attached files.

Reviewer #1: No

Reviewer #2: **Yes: **Serebe Gebrie

---

## [Author Response · Author response to Decision Letter 0]

5 Oct 2020

The responses to each comment of reviewers and editor are uploaded as a supporting file named "Response to reviewers".

---

## [Decision Letter · Decision Letter 1]

12 Oct 2020

Caregiver experience and perceived acceptability of a novel near point-of-care early infant HIV diagnostic test among caregivers enrolled in the PMTCT program, Myanmar: a qualitative study

PONE-D-20-22119R1

Dear Dr. Win Lei Yee,

We’re pleased to inform you that your manuscript has been judged scientifically suitable for publication and will be formally accepted for publication once it meets all outstanding technical requirements.

Kind regards,

Claudia Marotta

Academic Editor

PLOS ONE

Additional Editor Comments (optional):

Dear Authors congratulations

Reviewers' comments:

Reviewer's Responses to Questions

**Comments to the Author**

1. If the authors have adequately addressed your comments raised in a previous round of review and you feel that this manuscript is now acceptable for publication, you may indicate that here to bypass the “Comments to the Author” section, enter your conflict of interest statement in the “Confidential to Editor” section, and submit your "Accept" recommendation.

Reviewer #1: All comments have been addressed

Reviewer #2: All comments have been addressed

2. Is the manuscript technically sound, and do the data support the conclusions?

Reviewer #1: Yes

Reviewer #2: Yes

3. Has the statistical analysis been performed appropriately and rigorously? 

Reviewer #1: Yes

Reviewer #2: N/A

4. Have the authors made all data underlying the findings in their manuscript fully available?

Reviewer #1: Yes

Reviewer #2: Yes

5. Is the manuscript presented in an intelligible fashion and written in standard English?

Reviewer #1: Yes

Reviewer #2: Yes

6. Review Comments to the Author

Reviewer #1: Authors improve their manuscript and now can be accept in my opinion

The setting and the topic of study is very interesting

Reviewer #2: This study is novel and relevant to the study setting. All my comments and concerns are well addressed. Significant improvement has been made on the revised version. I feel the manuscript is technically and methodologically acceptable for publication.

7. PLOS authors have the option to publish the peer review history of their article (what does this mean?). If published, this will include your full peer review and any attached files.

Reviewer #1: No

Reviewer #2: **Yes: **Serebe Gebrie

---

## [Editor Report · Acceptance letter]

21 Oct 2020

PONE-D-20-22119R1 

Caregiver experience and perceived acceptability of a novel near point-of-care early infant HIV diagnostic test among caregivers enrolled in the PMTCT program, Myanmar: a qualitative study 

Dear Dr. Yee:

I'm pleased to inform you that your manuscript has been deemed suitable for publication in PLOS ONE. Congratulations! Your manuscript is now with our production department. 

Kind regards, 

on behalf of

Dr. Claudia Marotta 

Academic Editor

PLOS ONE